# ON TASK DESCRIPTION OF IN-CONTEXT LEARNING: A STUDY FROM INFORMATION PERSPECTIVE

## ABSTRACT

Transformers have demonstrated remarkable performance in a wide range of applications, making in-context learning an essential technique. Although the in-context learning has been widely applied, our understanding of its underlying processes still remains limited. In-context learning in transformers primarily relies on two types of information: in-context samples and task descriptions. While previous research has extensively investigated the influence of in-context samples on learning behavior, the role of task descriptions has not been adequately explored, despite their practical significance. In this paper, we present a study examining the impact of task descriptions on in-context learning performance of transformers. We devise a synthetic experiment setting, making the information of task description controllable. Through a series of well-designed experiments, we systematically vary task description information and assess the resulting effects on model performance across multiple tasks. Our findings reveal the complex roles of task descriptions: task descriptions will lead the model to ignore in-context examples; task descriptions will increase the lower bound of the in-context learning performance. This study contributes to a deeper understanding of the in-context learning mechanism in transformers, paving the way for more effective real-world applications of these powerful models.

## 1 INTRODUCTION

The impressive performance of transformers highlights the significance of in-context learning for real-world applications. In-context learning pertains to the Transformer's ability to learn from context-based prompts. This learning approach is utilized in numerous practical applications, including AI planning (Valmeekam et al., 2022; Xie et al., 2023), reasoning (Huang & Chang, 2022), image understanding (Alayrac et al., 2022) and autonomous agents (Wang et al., 2023), and can provide theoretical derivation for experimental results in other fields like cognitive science Sumers et al. (2023).

Despite the extensive use of in-context learning, our comprehension of its underlying mechanisms remains limited. Recent research has investigated in-context learning within a meta-learning framework (Gu et al., 2023; Min et al., 2021), offering insights into how Transformers utilize in-context demonstrations to tackle new tasks. However, Transformer employ in-context information in two ways: through in-context demonstrations and task descriptions. The role of task descriptions, though practically significant, has not been thoroughly examined. In this work, we adopt a different perspective by concentrating on how task descriptions influence in-context learning within a meta-learning framework.

The meta-learning framework (Gu et al., 2023; Min et al., 2021) is used to enrich in-context learning of Transformer, where the Transformer is directly trained to implement in-context learning. The task dataset for this framework is constructed by equations in the form of $(x \circ y) \mod p = r$, where $p$ is a prime number, $\circ$ represents for operators, and $r$ is the result of equation to be predicted. Under this framework, the prompt is formulated as $[\{(x_i, y_i, r_i)\}_{i=1}^{l}, (x_q, y_q)]$. $\{(x_i, y_i, r_i)\}_{i=1}^{l}$ can be regarded as few shot examples, while $x_q$ is the validation examples. The Transformer is expected to learn this task from the few show examples. This framework is also leveraged for exploration of in-context learning (Akyürek et al., 2022; Von Oswald et al., 2023; Garg et al., 2022; Chan et al., 2022a;b; Fu et al., 2023). Following previous studies, we also use this framework.

However, we are different in that the task description is given. That is, the prompt in our task is $[d, \{(x_i, y_i, r_i)\}_{i=1}^{l}, (x_q, y_q)]$, where $d$ denotes task description. To investigate the role of task description, we devise a synthetic experiment, where we can flexible control the complexity of the task description by assign the task description with different level of information. Specifically, given a task ground truth label $t$, we design task description $d$ to control the mutual information $I(t; d)$.

In the proposed experimental setup, we investigate the impact of task descriptions on in-context learning. Our findings are: $(i)$ task descriptions can divert model's attention in in-context examples, and this effect is related to the task description's information, and $(ii)$ task descriptions can raise the lower bound of in-context learning performance. Consequently, we observe a phase transition regarding the impact of task descriptions: those with insufficient information can impair in-context learning performance due to $(i)$, while task descriptions with abundant information can aid in-context learning due to $(ii)$. We find two cases where Transformers can achieve good in-context learning performance: 1) a large number of in-context examples with low-information task descriptions, and 2) high-information task descriptions. Additionally, we explore whether incorporating task prediction as an auxiliary task during training improves in-context learning performance. The results indicate that task prediction as a surrogate task benefits in-context learning in nearly all cases. To verify the generality of our findings, we conduct further studies on more realistic NLP tasks, which align with our experimental results on the synthetic tasks.

Our contributions can be summarized as

- The development of a new synthetic task for investigating the role of task description in in-context learning.

- The identification of a phase transition of the in-context learning performance when increasing the information of task description.

- The conduction of further research beyond synthetic tasks to corroborate the universality of our findings.

## 2 RELATED WORK

**In-context learning**  In recent years, the field of natural language processing (NLP) has witnessed significant advancements, particularly in the development of large-scale language models designed for in-context learning. These models, such as GPT-4 (OpenAI, 2023) by OpenAI, PaLM2 (Anil et al., 2023) by Google, and Llama (Touvron et al., 2023) by Facebook, have demonstrated remarkable capabilities to understand and generate human-like text by leveraging massive amounts of data and sophisticated algorithms. In-context learning refers to the model's ability to adapt its understanding and responses based on the specific context provided (Brown et al., 2020), which has been proven to be crucial in enhancing their performance across various NLP tasks, including AI planning (Valmeekam et al., 2022; Xie et al., 2023), reasoning (Huang & Chang, 2022), image understanding (Alayrac et al., 2022), and autonomous agents (Wang et al., 2023). However, despite the impressive progress, challenges remain in terms of the mechanism driving in-context learning. This paper focuses on understanding the mechanism of in-context learning from a synthetic tasks. The results make a further step towards understanding in-context learning from the aspect of task description.

**Exploration of in-context learning from synthetic tasks.**  Exploring in-context learning mechanisms in real applications poses a significant challenge due to the complexities and intricacies involved in practical scenarios (Min et al., 2022). Consequently, recent studies have shifted their focus towards understanding the mechanisms of in-context learning on specific synthetic tasks, which offer a more controlled environment for examining individual aspects of the learning process. For instance, linear regression tasks have been employed in several studies (Akyürek et al., 2022; Von Oswald et al., 2023; Garg et al., 2022) to delve into the in-context learning behavior of Transformer, while some researchers have turned their attention to image data to analyze the learning process. Moreover, investigations (Chan et al., 2022a;b; Fu et al., 2023) have been conducted from in-context and in-weights perspectives, examining the learning process through the lens of the model's internal representations and the role of weights. However, despite these valuable contributions, most explorations mentioned above tend to overlook the influence of task descriptions on the in-context learning process. Considering the practical significance of task descriptions in guiding

Transformer towards desired learning outcomes, it is essential to examine their impact on in-context learning performance to gain a more comprehensive understanding of the in-context learning mechanisms and improve the effectiveness of these powerful models in real-world applications.

**Task description in real in-context learning application.** In the realm of in-context learning, the prompt plays a crucial role in guiding the language model's response generation. A prompt is a textual input provided to the model, containing the necessary context and instructions that help the model understand the user's requirements and produce relevant responses. The task description in the prompt often includes specific questions, statements, or examples that outline the desired output, enabling the model to adapt and generate contextually appropriate text (Brown et al., 2020). The task description plays a important role in in-context learning by providing information about recognizing the task in real application (Pan, 2023; Cho et al., 2023). However, systematic studies about the role of task description and the mechanisms behind are lacking. This paper fills this gap by providing the analysis of task description under different situations.

## 3  FORMULATION AND MOTIVATION

We assume a dataset $\mathcal{D}$, comprising $N$ data samples $\mathcal{D} = \{x_i = (d_i, c_i, q_i, r_i, t_i)\}_{i=1}^N$, where $d_i$ denotes the task description for the $i$-th sample, and $c_i$ represents a sequence of task examples associated with $q_i$. For each data sample, given a query $q_i$, our objective is to predict the output of $q_i$ for task $t_i$, labeled as $r_i$. We partition the dataset into two subsets: $\mathcal{D}_{train}$ and $\mathcal{D}_{test}$. This partitioning should ensure that tasks in the test dataset remain unseen in the training dataset, i.e., for each task $i$ in the testing set $\mathcal{D}_{train}$, no $t_j$ exists in $\mathcal{D}_{test}$ such that $t_i = t_j$. The primary aim of in-context learning is to utilize the task description and examples for adapting the model, thereby optimizing its performance on previously unseen tasks. To accomplish this objective, we maximize the following function:

$$\mathbb{E}_{p(d,c,q)}\mathbb{E}_{q_\theta(r|d,c,q)} \log p(r|d,c,q). \tag{1}$$

Here $q_\theta(r|d, c, q)$ denotes the predicted distribution of target $r$, while p refers to real distribution. To analyze the aforementioned objective associated with task $t$, we employ the variational method, constructing an evidence lower bound. Given the intractable nature of the distribution $p(t|r, d, c, q)$, we approximate it using a parameterized distribution $q_\theta(t|d, c, q)$ as follows:

$$\begin{aligned}&\mathbb{KL}(q_\theta(t|d,c,q)|p(t|r,d,c,q))\\&= \mathbb{KL}(q_\theta(t|d,c,q)|p(t|d,c,q)) - \mathbb{E}_{q_\theta(t|d,c,q)} \log p(r|t,d,c,q) + \log p(r|d,c,q).\end{aligned} \tag{2}$$

Please refer to appendix A.1 for the proof. Considering the non-negative nature of the KL divergence, we can express the log-likelihood in the following manner:

$$\log p(r|d,c,q) \geq -\mathbb{KL}(q_\theta(t|d,c,q)|p(t|d,c,q)) + \mathbb{E}_{q_\theta(t|d,c,q)} \log p(r|t,d,c,q). \tag{3}$$

The first term signifies the task label prediction, whereas the subsequent term corresponds to the loss function employed in the in-context training for the GPT model. This equation, therefore, demonstrates that accurate task label prediction contributes to the maximization of the log-likelihood.

Incorporating the task description as a component of the input allows it to serve as a representation of the task itself. To assess the efficacy of this description, we examine encoder and decoder models that yield conditional distributions $q(d|t)$ and $p(t|d)$. Given that $q(t)$ embodies the marginal distribution of task $t$, we define the reconstruction error, denoted as $\mathcal{R}$, in the following manner:

$$\mathcal{R} = \mathbb{E}_{q(t)}\mathbb{E}_{q(d|t)}[-\log p(t|d)] \leq \mathbb{KL}(q(t,d)||p(t,d)) - I_q(t,d) + H_q(t). \tag{4}$$

Please see appendix A.2 for the proof. The aforementioned equation indicates that increasing the mutual information can reduce the negative log likelihood of $t$. The mutual information, denoted as $I_q(t, d)$, between task label $t$ and the task description $d$ can be formulated as follows:

$$0 \leq I(t; d) = \mathbb{E}_{p(t,d)} \left[\log \frac{q(t,d)}{q(t)q(d)}\right] = H_q(t) - H_q(t|d) \leq H_q(t). \tag{5}$$

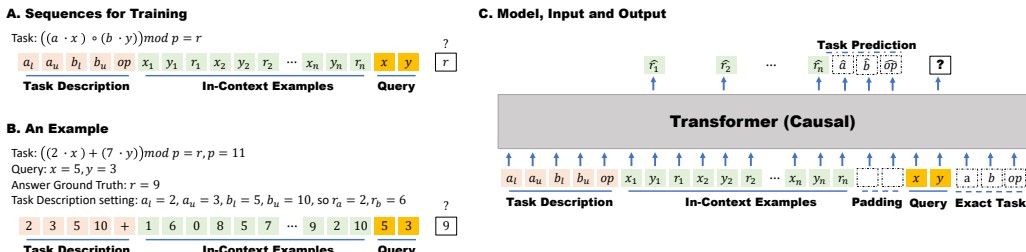

Figure 1: Experimental Setup. **A:** Our synthetic task dataset is constructed by simple equations. In training, the model will be given a sequence including task description, in-context examples and query. Only an inexact range of $a$ and $b$ will be implied in task description, and we train the model to calculate the answer $r$ of the operation given $x$ and $y$ as query. **B:** An example of input sequence. $r_a$ and $r_b$ are the inexact $ab$ ranges implied in task description. $r_a = a_u - a_l$, $r_b = b_u - b_l$,and $a_l, a_u, b_l, b_u$ stands for the possible lower and upper bounds of $a$ and $b$. **C:** Model, input and output. We use standard decoder-only Transformer, which takes a token sequence as input. The auto-regression is used to training the model. We calculate loss for the output sequence, and accuracy is calculated only on the answer of query equation. For task prediction, exact task $t = (a, b, op)$ will be added to the end of input token

Based on the aforementioned equation, we observe that the mutual information ranges from 0 to $H_q(t)$. Consequently, to examine the impact of mutual information, we propose incorporating its control in our experimental design. Please see Sec. 4 for the details.

In summary, we consider an in-context learning setting where the task is unseen in the training set. However, to simplify the problem, we assume that the task labels in the testing set are novel recombinations of the training ones. In order to reformulate the prediction into a compositional generalization problem, we derive a variational lower bound of the log likelihood as a new objective, as shown in Equation 3. The first term in it is for task prediction. Since we consider the task description as a representation of the task, the goodness of it has an impact on the model performance. By modeling it as a representation, we derive a quantity to estimate its goodness, as shown in Equation 4. Therefore, we design our experiments with some principles to analyze how to train our model for better in-context ability from the following perspectives: 1) the mutual information between the task description and the task; 2) with or without task prediction.

## 4 EXPERIMENTAL DESIGN

In this section, we will delve into the experimental design and its various components. We begin by outlining the design principles, which serve as the foundation for the entire experiment. With these principles in mind, the experimental design aims to study the factors impacting the model's in-context ability by a robust and flexible framework. Furthermore, this design allows for the future research on in-context learning, since it is a controllable benchmark for in-context learning.

**Design Principle** 1) **Controllable task description information**: The information provided in the task description can be directly manipulated, allowing for a precise control over the quantity of information presented to the model. 2) **Unseen evaluation tasks**: To ensure the model's ability to generalize, the evaluation tasks presented to the model are not included in the training data. This helps assess the model's performance in handling novel tasks. 3) **Information inference from multiple sources**: The model is designed to extract information of task from both the task description and in-context examples provided. This enables the model to adapt and learn from various sources of information.

### 4.1 TASK DESIGN

Our synthetic task dataset is constructed by equations in the form of $((a \cdot x) \circ (b \cdot y)) \bmod p = r$, where $p$ is a prime number, and $\circ$ can represent $+$, $-$ or $/$. For each task, $a$, $b$ and $\circ$ are randomly

selected and fixed, but only an inexact range of $a$ and $b$ will be implied in task descriptions, and we train the model to calculate the answer $r$ of the operation given $x$ and $y$ as query. Only half of available $ab$ pairs and $xy$ queries are seen in the training, and the remaining equations are used for evaluation. We choose $p = 11$ in all experiments.

The task description is given as $\langle a_l \rangle \langle a_u \rangle \langle b_l \rangle \langle b_u \rangle \langle op \rangle$, while $\langle a_l \rangle, \langle a_u \rangle, \langle b_l \rangle, \langle b_u \rangle$ stands for the possible lower and upper bounds of $a$ and $b$ separately, and $\langle op \rangle$ stands for the operator $+, -$ or $/$ used in this task. We change the given range of $a,b$ to control the quality of task description, and a larger $ab$ range refers to lower task description quality as more possible $ab$ pairs can be deduced. For a given task $((a \cdot x) \circ (b \cdot y)) \bmod p = r$, several examples are randomly selected and constructed as $< x_i, y_i, r_i >$, while $r_i = ((a \cdot x) \circ (b \cdot y)) \bmod p$.

## 4.2 Model and training

**Model**   For most experiments on synthetic tasks, we use a standard decoder-only causal Transformer (Vaswani et al., 2017) with 24 layers, an embedding length of 256, and 8 attention heads. For experiments on the natural language task CoFE (An et al., 2023), we follow their approach and use fine-tuned GPT2-Large as our model.

**Loss Function**   The auto-regression is used to train the model. Following GPT (Radford & Narasimhan, 2018), given a token sequence $x = (x_1, \ldots, x_T)$, we train the model to predict $p(x) = \prod_{t=1}^{T} p(x_t | x_{<t})$. We calculate loss for in-context examples, query, and the answer of query equation. The in-context examples are denoted as set $\mathcal{C}_{i-1}$. For $i > 1$, $\mathcal{C}_{i-1}$ represents for in-context example sequence $\{(x_1, y_1, r_1), \ldots, (x_{i-1}, y_{i-1}, r_{i-1})\}$. For $i = 1$, $\mathcal{C}_0$ is an empty. Specifically, we calculate the loss for the sequence $s = \{(x_1, y_1, r_1), \ldots, (x_L, y_L, r_L)\}$ and task description $d$ as follows:

$$\mathcal{L}(\theta, s, d) = \frac{1}{L} \sum_{i=1}^{L} l(f(\{d, \mathcal{C}_{i-1}, x_i, y_i\}), r_i), \tag{6}$$

where $l$ denotes the loss function, e.g., crossentropy loss is adopted in our setting. The task description is $d = (a_l, a_u, b_l, b_u, op)$. Accuracy is calculated only for the answer of query equation. For task prediction, task $t = (a, b, op)$ will be added to the end of input token, and loss for task prediction can be re-formulated as:

$$\mathcal{L}_t(\theta, s, d) = \frac{1}{L} \sum_{i=1}^{L} l(f(\{d, \mathcal{C}_{i-1}, x_i, y_i\}), r_i, t). \tag{7}$$

**Training configure**   We train the model for 200k steps, and use Adam optmizer with learning rate $1e^{-4}$ for all experiments. Minibatch size is set to 128 for training and validation on our synthetic tasks, and 4 for CoFE.

## 4.3 Impact Factors in Prompt

**Task description**   We leverage the mutual information to evaluate the task description. Since only inexact ranges of $a$ and $b$ are implied in task description as $r_a = a_u - a_l$ and $r_b = b_u - b_l$, the quality of task description can be controlled and quantified by changing $r_a$ and $r_b$. To be specific, suppose the full number of available $ab$ pairs is $n_{ab}$, and the inexact $ab$ range implied in task description are $r_a$ and $r_b$. Then, given this task description, we can narrow down possible $ab$ pair numbers from $n_{ab}$ to $r_a \cdot r_b$. This indicates that the information gain given by the task description is $log(n_{ab}/(r_a \cdot r_b))$

**Number of Examples**   We use the number of examples to control the information conveyed by demonstration. For a given task, adding more in-context examples refers to providing more information by demonstration.

# 5 EXPERIMENTS RESULTS

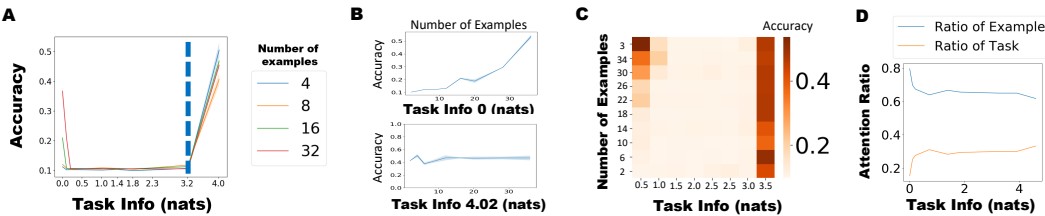

Figure 2: Phase Transition when increasing the information of task description.Shaded areas indicates +/- variance. **A:** The task description will distract in-context learning ability of transformer when its information is less than a threshold, while it will improve in-context learning after that. **B:** Before the Phase Transition, the number of in-context examples significantly impacts in-context learning, while after that, it has almost no influence. **C:** The model can obtain in-context learning only under two cases: 1) low info under large number of in-context examples. 2) High info task description. **D:**Attention explanation. The ratio of in-context examples in attention keeps declining with more task description information.The task description will divert the model's attention in in-context examples.

## 5.1 HOW DOES TASK DESCRIPTION IMPACT IN-CONTEXT LEARNING

We use the accuracy of the predicted results of query examples to reflect in-context learning performance, and use the mean of five runs to reduce the randomness. The results are presented in Figure 2. Our main findings are as following:

**A Phase Transition course can be observed.** Figure 2A depicts the variation of accuracy with the amount of information and the number of in-context examples. Before a certain information threshold, the accuracy remains at a low level. At this stage, significant accuracy gain can only be observed when more in-context examples are added. However, after this information threshold, the accuracy grows rapidly with information gain, but keeps relatively stable with changes in the number of in-context examples.

**Before the Phase Transition, the task description will distract in-context learning ability of transformer, but will improve in-context learning after that.** Figure 2B gives a clearer demonstration of Phase Transition. The accuracy grows as the number of in-context examples increases before Phase Transition, but stays relatively constant within a large range of in-context example numbers after Phase Transition.

**Phase Transition course leads to two in-context learning stage of transformer.** As shown in Figure 2C. The model can achieve a high accuracy only when given low-information task description under large number of in-context examples, or given high-information task description.

## 5.2 THE PHASE TRANSITION OF TASK DESCRIPTION.

In the previous section, we discover the phase transition of task description. Here, we further investigate the reason behind it. Specifically, we infer the possible reasons from the follow two perspectives:

**The task description will lead the model to ignore the information from in-context examples.** We calculate the ratio of in-context examples and task description in transformer attention, given same input sequence. As shown in Figure 2D, the ratio of in-context examples in attention keeps declining with more task description information. On the contrary, the attention ratio of task description increases when more task-related information are given. This indicates that adding task description info will divert model's attention in in-context examples.

Figure 3: Results of task prediction. **A:** A demonstration of accuracy gain (Predicting tasks v.s. without predicting tasks). Acc(p.t.) refers to accuracy on predicting results under predicting tasks setting, Acc(w/o p.t.) refers to corresponding accuracy without task prediction. Accuracy gain means the value of Acc(p.t.) - Acc(w/o p.t.). Using task prediction as proxy task can significantly improve in-context learning ability of Transformer. **B:** Task accuracy increases with task description info. **C:** The number of in-context examples can impact task prediction accuracy only under low info task description. **D:** Task info have greater influence than the number of examples.

**Higher information of task description will increase the lower bound of performance.** As illustrated in Eq 3, higher mutual information signifies that the task description is a good representation of the actual task. In other words, the task description captures the essential aspects and the underlying structure of the task, providing the model with valuable insights and a more accurate understanding of the problem it needs to solve. When the mutual information is high, it means that knowing the task description reduces the uncertainty about the prediction of task itself. Consequently, when the task description has high mutual information with the task, the model can leverage this strong representation to make better decisions and predictions, even when faced with limited or ambiguous examples.

To study how predicting task label impacts the performance of in-context learning (measured using the accuracy of validation query examples), we conduct experiments by adding an extra loss between the predicted task label and ground truth task label. By comparing the gain (with predicting task label v.s w/o predicting task label), we can evaluate the impact of task prediction.

**Predicting the task can improve in-context learning performance.** The results are presented in Figure 3A. A warm color in Figure 3A refers to positive accuracy gain. A performance improvement can be observed under different task descriptions and in-context example settings, as the points in Figure 3A are mainly colored warm. And the accuracy gain increases sharply with mutual info, at a similar threshold with that in Figure 2A, demonstrating a phase transition for the accuracy gain. Before Phase Transition, such accuracy gain tends to grow with the number of in-context examples. There are some cases where the performance slightly drops due to randomness. After Phase Transition, the accuracy gain remains significant and stable.

The performance of task label prediction can also reflect whether the model understand what the task is. Besides the accuracy of query examples, we further examine the accuracy of the predicted task label (denoted as task accuracy for simple). As shown in Figure 3B and Figure 3C, the model can predict tasks better when given more task description information or more in-context examples. Figure 3C depicts that the number of in-context examples has an obvious impact on task prediction accuracy only under low info task descriptions. According to Figure 3D, increasing both task description information and the number of in-context examples can enhance the model's ability in task prediction, but the influence of task description is relatively more significant.

## 5.3 BEYOND THE SYNTHETIC EXPERIMENT

To verify that the discovery from the synthetic experiment also hold on the real task, we conduct another experiment on the more realistic task on a realistic natural language dataset.

We experiment on CoFE (An et al., 2023), a natural language dataset for compositional generalization. The training set covers all the primitives while lacking certain combinations, this enforces the model to understand and re-combine known components in language. We select 3 categories of combinations of primitives in the dataset: Primitive Substitution, Primitive Structural Alternation

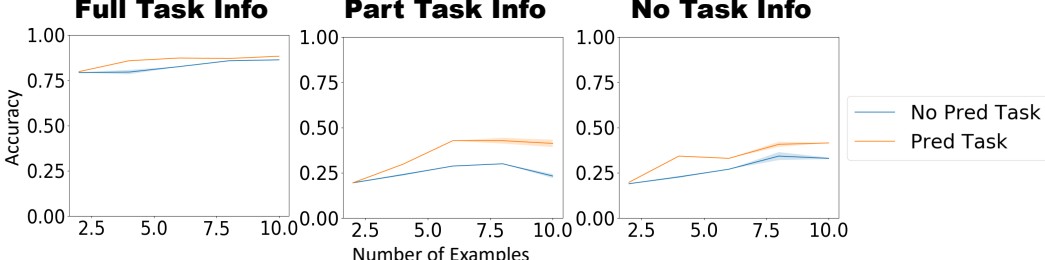

Figure 4: Experiments on real tasks. We design three different settings of task description. In **Full Task Info** experiment, all task information are given. In **Part Task Info** experiment, the info of target primitive is excluded. In **No Task Info** experiment, no task description is added. We experiment on all three info settings given 2, 4, 6, 8, 10 in-context examples separately. We find that the conclusions of experiments of synthetic tasks are also held in real tasks

and Phrase Recombination. The model is trained to predict 4 types of primitives for each combination category, resulting in 12 tasks. In our experiment, the training set consists of 4 randomly selected tasks, covering all 4 types of target primitives and all 3 combination categories. The test set consists of the remaining 8 tasks. Examples of data in CoFE are provided in the appendix.

We design three settings of task description containing different amount of information. All task information are given in Full Task Info experiment. In Part Task Info experiment, we only imply the combination category of the task in task description, but leave out the type info of the target primitive. In No Task Info experiment, no task description is added. We experiment on all three info settings under different numbers of in-context examples. The results are given in Figure 4.

**The conclusions of synthetic experiments are still held.** In all three settings, using task prediction as proxy task can significantly improve accuracy, confirming the impact of task prediction on model's in-context learning ability. Figure 4A depicts that experiments on Full Task Info achieve the highest accuracy across all settings. This indicates that when given high info task descriptions, the model can obtain higher in-context learning ability than given low info. However, when given incomplete and limited task information, as shown in Figure 4B, the model achieve relatively low accuracy and obtains limited accuracy gain with an increasing number of in-context examples. The results demonstrate that low info task descriptions mislead in-context learning. Those observations are well-aligned with the findings on the above synthetic experiment, indicating our findings on synthetic experiments can be well scale to real word cases.

### 5.4 ABLATIONS

**No task description during training.** We present the model's accuracy given no task description and different number of in-context examples. It can be depicted in Table 1 that the accuracy grows with in-context example number. This table actually refers to zero mutual information in Figure 2A and Figure 2C, and it can be inferred from Figure 2 that model given full info task description always outperforms model given zero task info.

**No in-context examples during training.** Table 2 lists the model's accuracy given different amount of task info and no in-context examples. When given maximal info (4.6052, referring to totally accurate task description), the model can achieve 0.8641 accuracy, better than all other info level settings, but fall behind models given both full task description and in-context examples. This infers model's ability in understanding task description. Also, it can be seen that under no example setting, the accuracy grows with information gain. The growing trend is relatively tiny given low task info, but speeds up when more task info added. Such performance pattern keeps align with experiments given both task description and in-context examples.

| Task Info (nats) | 0 | 0.21 | 0.4462 | 0.7133 | 1.0217 | 1.609 | 2.3026 | 3.2189 | 3.6243 | 3.912 | 4.3175 | 4.6052 |
|---|---|---|---|---|---|---|---|---|---|---|---|---|
| Accuracy | 0.1017 | 0.1027 | 0.1036 | 0.1041 | 0.1038 | 0.1053 | 0.1083 | 0.1089 | 0.2104 | 0.2834 | 0.4267 | 0.8641 |

Table 1: Ablation Experiments: No in-context example and different amount of task information.

| Number of In-context Examples | 0 | 4 | 8 | 12 | 16 | 24 | 32 | 36 |
|---|---|---|---|---|---|---|---|---|
| Accuracy | 0.1017 | 0.1117 | 0.1234 | 0.1320 | 0.2094 | 0.2955 | 0.3670 | 0.5367 |

Table 2: Ablation Experiments: No task info and different numbers of in-context examples.

## 6 LIMITATION

A potential limitation of this work lies in the synthetic experimental setting that has been employed to investigate the impact of task descriptions on in-context learning performance of Transformers. While this approach enables the systematic exploration of task description information and its influence on model performance, it may not fully capture the nuances and challenges encountered in real-world scenarios. The simplification and controlled nature of the synthetic setting might result in findings that do not entirely generalize to practical applications, where language models have to deal with diverse tasks, more complex instructions, and ambiguous or incomplete information.

Moreover, the study's focus on task descriptions may not comprehensively address other factors that could significantly influence the performance of Transformers, such as the quality and representativeness of training data, model architecture, or the fine-tuning process. In the pursuit of a deeper understanding of in-context learning, it is essential to consider these additional elements to ensure a more holistic perspective on the behavior and performance of Transformers in real-world applications.

## 7 CONCLUSION

In conclusion, transformers have exhibited exceptional performance in various applications, with in-context learning emerging as a vital technique in the field. Despite its widespread use, our comprehension of the underlying mechanisms of in-context learning remains limited. This study delves into the crucial yet underexplored role of task descriptions in in-context learning performance, shedding light on their impact on transformers. By conducting a series of well-designed experiments in a synthetic setting, the research systematically investigates the influence of task description information on model performance across diverse tasks and domains. The results underscore the importance of task descriptions as a guiding factor for transformers to achieve desired learning outcomes. The well-designed experiments conducted in a synthetic setting highlight the need for carefully crafting task descriptions to enhance model performance and generalization because of the phase transition. Ultimately, this study deepens our understanding of the in-context learning processes in transformers and lays the foundation for more efficient and effective real-world applications of these advanced models.

However, it is crucial to acknowledge the limitations of the synthetic experimental setting and consider the additional factors that may influence transformer performance in real-world scenarios. While this study sheds light on the impact of task descriptions, future work should address the various challenges and complexities that transformers face in practical applications, such as diverse tasks, ambiguous instructions, and incomplete information.

In future work, several avenues can be pursued to further advance our understanding of in-context learning on task description in Transformers and enhance their practical applications. For example, it is valuable to explore the development of automated methods for generating optimal task descriptions, which could alleviate the challenges in crafting effective prompts and improve model performance across a range of tasks. Secondly, investigating the impact of incorporating more structured or hierarchical task descriptions could provide valuable insights into the model's ability to understand complex instructions and generate more contextually appropriate responses.

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

# A  APPENDIX

## A.1  THE DERIVATION OF EQUATION 3

In the following two sections, we have primarily drawn upon the proofs VAE (Kingma & Welling, 2013) and in Belghazi et al. (2018) as key literature sources to prove our claims.

Using Bayes rule, we can obtain the following derivation:

$$
\begin{aligned}
&\mathbb{KL}(q_\theta(t|d,c,q)|p(t|r,d,c,q)) \\
&= \mathbb{E}_{q_\theta(t|d,c,q)}[\log q_\theta(t|d,c,q) - \log p(t|r,d,c,q)] \\
&= \mathbb{E}_{q_\theta(t|d,c,q)}\left[\log q_\theta(t|d,c,q) - \log \frac{p(r|t,d,c,q)p(t|d,c,q)}{p(r|d,c,q)}\right] \\
&= \mathbb{E}_{q_\theta(t|d,c,q)}\left[\log \frac{q_\theta(t|d,c,q)}{p(t|d,c,q)} - \log p(r|t,d,c,q)\right] + \log p(r|d,c,q) \\
&= \mathbb{KL}(q_\theta(t|d,c,q)|p(t|d,c,q)) - \mathbb{E}_{q_\theta(t|d,c,q)}\log p(r|t,d,c,q) + \log p(r|d,c,q)
\end{aligned}
\tag{8}
$$

## A.2  THE DERIVATION OF EQUATION 4

We can rewrite the reconstruction error with the conditional distribution $p(t|d) = p(t,d)/p(d)$:

$$
\begin{aligned}
\mathcal{R} &= \mathbb{E}_{q(t)}\mathbb{E}_{q(d|t)}[-\log p(t|d)] = \mathbb{E}_{q(t,d)}\left[\log \frac{q(t,d)}{p(t,d)}\right] - \mathbb{E}_{q(t,d)}[\log q(t,d)] + \mathbb{E}_{q(d)}[\log p(d)] \\
&= \mathbb{KL}(q(t,d)|p(t,d)) - \mathbb{E}_{q(t,d)}[\log q(t,d)] + \mathbb{E}_{q(d)}[\log p(d)],
\end{aligned}
\tag{9}
$$

where the first term is KL divergence, the second term is the joint entropy $H_q(t,d)$. We focus on the third term:

$$
\mathbb{E}_{q(d)}[\log p(d)] = \mathbb{E}_{q(d)}[\log \frac{p(d)}{q(d)}] + \mathbb{E}_{q(d)}[\log q(d)] = -\mathbb{KL}(q(d)|p(d)) + H_q(d)
\tag{10}
$$

We bring Eq. 10 into Eq. 9, then the joint entropy and entropy can be formulated as:

$$
\begin{aligned}
-\mathbb{E}_{q(t,d)}[\log q(t,d)] + H_q(d) &= -\mathbb{E}_{q(t,d)}\left[\log \frac{q(t,d)}{q(t)q(d)}\right] + \mathbb{E}_{q(t)}[\log q(t)] \\
&= -I_q(t;d) + H_q(t)
\end{aligned}
\tag{11}
$$

Since the KL-divergence is non-negative, we obtain the bound:

$$
\begin{aligned}
\mathcal{R} &= \mathbb{KL}(q(t,d)|p(t,d)) - \mathbb{KL}(q(d)|p(d)) - I_q(t;d) + H_q(t) \\
&\leq \mathbb{KL}(q(t,d)|p(t,d)) - I_q(t;d) + H_q(t)
\end{aligned}
\tag{12}
$$

## A.3  SUPPLEMENTARY DESCRIPTION FOR FIGURE 2

**Figure 2A.** Figure 2A and its numeral results (Table 3) depicts the variation of accuracy with the amount of information and the number of in-context examples. Before a certain information threshold (about 3.2 nats), the accuracy remains at a low level. At this stage, significant accuracy gain can

| Task Info(nats) | 0 | 0.21 | 0.45 | 0.73 | 1.02 | 1.39 | 1.61 | 1.83 | 2.30 | 3.22 | 3.40 | 3.91 | 4.02 | 4.27 | 4.61 |
|---|---|---|---|---|---|---|---|---|---|---|---|---|---|---|---|
| Accuracy(0 ex) | 0.1017 | 0.1020 | 0.1027 | 0.1030 | 0.1034 | 0.1030 | 0.1042 | 0.1048 | 0.1062 | 0.1119 | 0.1164 | 0.2104 | 0.2834 | 0.4267 | 0.8641 |
| Accuracy(4 ex) | 0.1118 | 0.1071 | 0.1062 | 0.1036 | 0.1023 | 0.1020 | 0.1030 | 0.1062 | 0.1083 | 0.1174 | 0.1202 | 0.2216 | 0.5052 | 0.5877 | 0.9323 |
| Accuracy(32 ex) | 0.3670 | 0.2083 | 0.1670 | 0.1046 | 0.1062 | 0.1070 | 0.1084 | 0.1097 | 0.1101 | 0.1211 | 0.1314 | 0.2137 | 0.4548 | 0.7926 | 0.9688 |

Table 3: Numerical Results in Figure 2A. Val Accuracy given different task info and number of in-context examples. Accuracy(0 ex), Accuracy(4 ex), Accuracy(32 ex) refer to val accuracy of models trained with no in-context example, 4 in-context examples, 32 in-context examples, respectively.

| Number of Examples | 0 | 2 | 4 | 8 | 12 | 16 | 20 | 24 | 28 | 32 | 36 |
|---|---|---|---|---|---|---|---|---|---|---|---|
| Accuracy (Mutual Info 0 nats) | 0.1017 | 0.1052 | 0.1117 | 0.1234 | 0.1320 | 0.2094 | 0.1876 | 0.2414 | 0.2955 | 0.3670 | 0.5367 |
| Accuracy (Mutual Info 4.02 nats) | 0.3530 | 0.4267 | 0.5052 | 0.4067 | 0.4695 | 0.4687 | 0.4656 | 0.4573 | 0.4678 | 0.4548 | 0.4674 |

Table 4: Numerical Results in Figure 2B. Val Accuracy given different numbers of in-context examples before and after phase transition threshold.

only be observed when more in-context examples are added. However, after this information threshold, the accuracy grows rapidly with information gain, but keeps relatively stable with changes in the number of in-context examples. Some digital results are listed in Table 3, and we choose some special settings: no in-context examples given, few in-context examples given(Accuracy(4 ex)), and plentiful in-context examples given(Accuracy(32 ex)). Under all settings, accuracy basically grows with mutual info gain. However, before a certain information threshold (around 3.2 nats concerning results sampled in Table 3), the accuracy gain remains relatively subtle to be 0.001 between neighbouring info settings.

For clarity and readability, not all sampled info settings are labeled in x axis of Figure 2A. As shown in Table 3, we actually sampled 15 info points given same number of in-context examples.The minimal mutual info given by task description is 0, referring to no valid information added by task description. The maximal possible mutual task info, under our experiment setting, is 4.61 nats. The mutual information we utilized here is formulated by $ln(n_{ab}/(r_a \cdot r_b))$, $r_a = a_u - a_l$ and $r_b = b_u - b_l$ stand for the inexact ranges of $a$ and $b$ are implied in task description, and $n_{ab}$ stands for the full number of available $ab$ pairs. Given maximal upper bound$(a_u, b_u)$ 10 and minimal lower bound$(a_l, b_l)$ 1, there can be 100 different $r_a, r_b$ range pairs if $ab$ are restricted to integers. Thus, 100 levels of task info can be experimented, resulting in maximal info 4.61 nats. We sample several mutual info settings, trying to keep uniform sampling intervals in task info.

We perform 5-fold experiments and the variances are given as shaded areas. Only after information threshold (around 4 nats), the variances can be obviously observed in figure.

**Figure 2B.** It can be seen in Fig 2B and its numeral results (Table 4) that before the threshold (task info less than 3.2 nats), the number of in-context examples significantly impact in-context learning, while after that, it has almost no influence. And the lower sub-figure in Fig 2B corresponding to the right most info point in Fig 2A( mutual Info 4.02 nats), which clearly depicts that the accuracy stays relatively constant within a large range of in-context example numbers.

## A.4  CoFE dataset

CoFE dataset (An et al., 2023) is constructed based on COGS, a compositional generalization benchmark designed for the fine-tuning paradigm. Here, compositional generalization refers to understanding and producing novel expressions by recombining known components in language, and is an important human ability. COGS, as well as CoFE, are designed for semantic parsing tasks. In these datasets, the training set covers all the primitives but lacks certain combinations, and the test set is made up of these missing combinations, so the model has to learn to translate natural language expressions into semantic representations.

The combinations in CoFE can be divided into five categories: Primitive Substitution (Compose a primitive with a grammatical role), Primitive Structural Alternation (Compose a primitive with a sentence structure (e.g., "subj. verb obj."), Phrase Recombination (Compose a prepositional phrase with a grammatical role), Longer Chain and Deeper Nesting. We only employ the former three combinations in our experiments, since the latter two are not suitable for our mutual task info setting.

| Category | In-context Examples | Test Case |
|---|---|---|
| Primitive Substitution | input:shark
output:NONE(SHARK,NONE,NONE)
input:A girl grew the boy.
output:DRAW(Girl,BOY,NONE) | input:The shark drew a boy.
output:DRAW(SHARK,BOY,NONE) |
| Primitive Structural Alternation | input:The goose baked.
output:BAKE(GOOSE,NONE,NONE)
input:A teacher noticed a chicken.
output:NOTICE(TEACHER,CHICKEN,NONE) | input:A teacher baked the chicken.
output:BAKE(TEACHER,CHICKEN,NONE) |
| Phrase Recombination | input:Logan mailed Stella the cake in the pile.
output:MAIL(LOGAN,IN,STELLA)
input: The goose rolled a baby in a room.
output:ROLL(GOOSE,IN,NONE) | input:A visitor in the pile rolled a resident.
output:ROLL(IN,RESIDENT,NONE) |

Table 5: *
Examples of data in CoFE.

Some data examples are given in Table 5. The three combination categories( or main tasks) refer to different forms of language composition, while the output are corresponding primitives like subject, verb or object. For each main task, four types of primitives are predicted, and can be split into four sub tasks, resulting in 12 different tasks. The model is trained to predict 4 types of primitives for each combination category, resulting in 12 tasks. In our experiment, the training set consists of 4 randomly selected tasks, covering all 4 types of target primitives and all 3 combination categories. The test set consists of the remaining 8 tasks.

We design three settings of task description containing different amount of information. All task information are given in Full Task Info experiment. In Part Task Info experiment, we only imply the combination category of the task in task description, but leave out the type info of the target primitive. In No Task Info experiment, no task description is added. We experiment on all three info settings under different numbers of in-context examples. In task description, the combination categories are tokenized as 1,2,3 and the target primitive type are denoted as 11-14. All words in CoFE are tokenized starting from 100 to avoid messing up with task description.

