# OpenReview forum: "On Task Description of In-context Learning: A Study from Information Perspective"
_ICLR.cc/2024/Conference — Submitted to ICLR 2024_

### Official Review · Reviewer_9ENX · 2023-10-27

**Soundness:** 1 poor
**Presentation:** 2 fair
**Contribution:** 1 poor
**Rating:** 3
**Confidence:** 4

**Summary:**

The authors study how task descriptions affect the in-context learning abilities of transformer models. To do so, they devise a synthetic experiment setting in which the information provided by the task description is controllable. They find that task descriptions can suppress the model of learning from examples and that they can increase the lower bound of in-context learning performance.

**Strengths:**

The proposed task and experimental setup were clean.

How task descriptions affect in-context LLMs would indeed be a worthy question to study.

**Weaknesses:**

The authors claim (at multiple points in the text) that their work “contributes to a deeper understanding of the in-context learning mechanism in LLMs.” However, this is not true as none of the experiments involves an LLM. Instead, their results only study in-context learning in a transformer-based model. The claim of improving our understanding of LLMs is misleading.

Furthermore, the authors state that our understanding of in-context learning in the meta-learning setting (which they study) is limited. I don’t think that this is true. We have a pretty good theoretical understanding of in-context learning in meta-learned neural networks: they implement (under ideal conditions) Bayesian inference for the distribution of tasks they were trained on (Ortega et al. 2019). Given this correspondence, I don’t find the main results from the present paper surprising. If the mutual information between the task and its description is high, no hidden variable has to be inferred, leading to high performance and no learning from samples. If it is low, the hidden variables have to be inferred to make good predictions, and that is easier with more samples.

Ortega, Pedro A., et al. "Meta-learning of sequential strategies." arXiv preprint arXiv:1905.03030 (2019).

The notation is all over the place. This starts with the equations on page 1 for which the meaning of y and r is not provided. Letters are used inconsistently. q_{\theta} is not defined. Figure 1 suddenly uses r to denote targets. In general, given these issues, I had a hard time following the description provided in the formulation and motivation section, even though I think I understood the method from the general framing.

Important details about the training task distribution are missing. How are the parameters sampled?

Minor:

There seems to be a leftover from editing: "There are several grammar errors and issues with the clarity in the given passage. I will provide a corrected version below, with changes and suggestions highlighted in brackets."

Figure 3A:
* no axis labels, and hence impossible to interpret.
* mentioned after Figure 3B in the text.

The writing is ungrammatical in places and should be double-checked. Examples:
* The conclusions of synthetic experiments are still held.
* Those observation is well align with the findings on the above synthetic experiment.
* As the task label prediction can also reflect whether the model understand what the task is.

**Questions:**

In the introduction, the authors claim that task descriptions with minimal information can impair in-context learning performance because they hinder the model’s ability to learn from in-context examples. I don’t think that this is correct given the presented results (Figure 1).

Figure 2: I don’t understand why performance for medium task information is worse than performance for low task information. Shouldn’t more information always be better?

---

> ### Author Response · Authors · 2023-11-22
> **Rebuttal by Authors-part1**
>
> Thank you for your valuable comments and suggestions and the encouraging words. We appreciate the time and effort you took to review our paper. Please find below our responses to your concerns and the changes we have made to address them.
>
> $\bullet$ Thank you for your valuable feedback. We are regret that our writing is not clear and can be misleading, and we have revisited and revised the paper for clarity.
>
> $\bullet$ We would like to clarify that the word "limited"  here refers to that relatively less exploration has been spent on the impact of task descriptions on in-context learning performance, especially compared to extensive investigation on the influence of in-context samples, such as the works Ortega et al.[1]. We are regret that our writing has led to your misunderstanding, and promise to modify the corresponding presentation in the paper.
>
> [1] Ortega, Pedro A., et al. "Meta-learning of sequential strategies." arXiv preprint arXiv:1905.03030 (2019).
>
> $\bullet$ Thank you for your detailed feedback. We are regret that our notation is not clear enough. We have thoroughly revised and polished the paper for clarity and readability. We have added the meaning of y and r in intro section (highlighted in red). $q_theta$ is the predicted distribution of target $r$, and we have added the definition in method section (highlighted in red) and unified the notation of targets as r.
>
> $\bullet$ Thank you for questions regarding parameter sampling.
> We adopt the same parameter sampling with GROKKING[2], parameters are sampled uniformly, which makes the calculation of mutual information easy.
>
> [2] Alethea Power, Yuri Burda, Harri Edwards, Igor Babuschkin, and Vedant Misra. Grokking: Generalization beyond overfitting on small al-gorithmic datasets, 2022

---

> ### Author Response · Authors · 2023-11-22
> **Rebuttal by Authors-part2**
>
> $\textbf{Q1}$: Thank you for your feedback. We are regret our writing are not clear enough and lead to your confusion. We have revised corresponding sections in the paper for clarity, and we would like to clarify our results and conclusion as follows:
>
> | Task Info(nats) | 0 | 0.21 | 0.45 | 0.73 | 1.02 | 1.39 | 1.61 | 1.83 | 2.30 | 3.22 | 3.40 | 3.91 | 4.02 | 4.27 | 4.61 |
> | ------ | ------ | ------ | ------ | ------ | ------ | ------ | ------ | ------ | ------ | ------ | ------ | ------ | ------ | ------ | ------ |
> |Accuracy(0 ex) | 0.1017 | 0.1020 | 0.1027 | 0.1030 | 0.1034 | 0.1030 | 0.1042 | 0.1048 | 0.1062 | 0.1119 | 0.1164 | 0.2104 | 0.2834| 0.4267 | 0.8641 |
> |Accuracy(4 ex) | 0.1118 | 0.1071 | 0.1062 | 0.1036 |  0.1023 | 0.1020 | 0.1030 | 0.1062 | 0.1083 | 0.1174 | 0.1202 | 0.2216 | 0.5052 |0.5877 | 0.9323 |
> |Accuracy(32 ex) | 0.3670 | 0.2083 | 0.1670 | 0.1046 | 0.1062 | 0.1070 | 0.1084 | 0.1097 | 0.1101 | 0.1211 | 0.1314 | 0.2137 | 0.4548 | 0.7926 | 0.9688 |
>
> $\hspace{2em}\textbf{1. Experimental Results: given insufficient task info, the accuracy gain with added in-context examples is subtle.}$ Table above gives detailed results of Figure 2A.
> The results of 0, 4 and 32 in-context examples are provides.
> When given insufficient task info (0.73 to 3.22), the performance gain is rather subtle, compared to the one with task info less than 0.73 or larger than 3.22.
> This also indicates the insufficient task info may mislead the model to ignore the learning of information from in-context examples.
>
> $\hspace{2em}\textbf{2. Experimental Results: given insufficient task info, the accuracy is low.}$ As shown in Fig 2A and Table above, for each row in the table,
> before a certain information threshold (about 3.2 nats), the accuracy remains at a relatively low level.
> Only after this information threshold, the accuracy grows rapidly as the task information gains.
> Although accuracy basically grows with mutual info gain, such accuracy gain remains relatively subtle to be 0.001 before the certain information threshold (around 3.2 nats), which is much smaller than a linear speed. In this sense, we say that: insufficient information can impair in-context learning performance.
>
> $\hspace{2em}\textbf{3. Explanation: model's attention in in-context examples is diverted by task description.}$ We explore the reason for model's relatively low accuracy given insufficient task info, and find that the task description may lead the model to ignore the information from in-context examples. As depicted in Fig 3B, the ratio of in-context examples in attention keeps declining with more task description information. On the contrary, the attention ratio of task description increases when more task-related information are given. This indicates that adding task description info will divert model's attention in in-context examples, hindering the model’s ability to learn from in-context examples.
>
> $\textbf{Q2}$: Thank you for your question regarding the experiment results. We are regret our writing has led to your confusion. We would like to provide a detailed explanation for the phenomenon that performance for medium task information is worse than performance for low task information, under some settings.
>
> $\hspace{2em}\textbf{1. Without in-context examples, the performance for medium task information is better than performance for low task information.}$
> As illustrated in Table above, Accuracy(0 example) setting, referring to experiments given no in-context examples, the accuracy grows accordingly with information gain. The growing trend is relatively tiny given low task info, but speeds up when more task info added.
>
> $\hspace{2em}\textbf{2. Task description diverts model's attention in in-context examples, but at varying levels considering the amount of task info. }$
> When more in-context examples are added, as Accuracy(4 examples) setting and Accuracy(32 examples) setting show, the general model performance is improved, but at varying levels. For medium task info settings, as depicted in Figure 2D, the model's attention in in-context examples is diverted by insufficient task info, which hinders the performance gain. However, the distraction of low info task description is relatively insignificant, resulting in seemingly good performance.

---

> ### Author Response · Authors · 2023-11-22
> **Rebuttal by Authors-part3**
>
> $\textbf{Our results and contributions}$: Thank you for your insightful feedback. You are correct that many works have extensively investigated  in-context learning in meta-learned neural networks. However, our primary contribution lies in how \textbf{task descriptions} influence in-context learning within a meta-learning framework, whose role has not been adequately explored by previous research. We would like to highlight our results and contributions as follows:
>
> $\hspace{1em}\textbf{We design a synthetic experiment to make task-description controllable and quantifiable.}$
> We devise a synthetic experiment setting, making the information of task description controllable. Through a series of well-designed experiments, we systematically vary task description information and assess the resulting effects on model performance across multiple tasks.
>
> $\hspace{1em}\textbf{We observe several meaningful experimental results.}$
> We observe a phase transition regarding the impact of task descriptions: those with insufficient information can impair in-context learning performance, while task descriptions with abundant information can aid in-context learning. Additionally, we explore whether incorporating task prediction as a auxiliary task during training improves in-context learning performance, and provide theoretical proof. The results indicate that task prediction as a surrogate task benefits in-context learning in nearly all cases.
>
> $\hspace{1em}\textbf{We have verified our findings on realistic tasks.}$
> To verify the generality of our findings, we conduct further studies on more realistic NLP tasks, which align with our experimental results on the synthetic tasks.
>
> As according to $\textbf{Reviewer hvVV}$, we are working on a "very timely problem". $ \textbf{Reviewer YT61}$ also "like the problem of how task information impacts ICL". And our synthetic experiment is appreciated by $\textbf{Reviewer hvVV}$, $\textbf{Reviewer YT61}$ and $\textbf{Reviewer DYTw}$, being "a refreshingly different approach to yet another heuristic and ad-hoc method for improving in-context learning in an intransparent fashion".
>
> We have revised our paper to correct our inappropriate expression and clarify our ideas, results and contributions. Figures are refined and notations are unified. We appreciate your comments and sincerely hope for rereading.

---

> > ### Comment · Reviewer_9ENX · 2023-11-22
> >
> > Thanks to the authors for their reponse. I will not have the time to go through it today. I'll read it later this week and update here again if possible.

---

### Official Review · Reviewer_DYTw · 2023-10-31

**Soundness:** 4 excellent
**Presentation:** 2 fair
**Contribution:** 3 good
**Rating:** 8
**Confidence:** 4

**Summary:**

This work explores the relationship between a task description and number of in-context examples, and particularly its effect on in-context learning. They implemented a synthetic task that involves solving algebraic equations and trained a standard transformer architecture on this synthetic dataset in which each task example contains both a task description, in-context examples, and a query example with which to evaluate on. Since its a controlled, synthetic task, they were able to control the amount of information about the task the task description will supply. They found an inflection point in which, at a certain threshold of task description information, the accuracy will dramatically rise with respect to the amount of information in the description. Additionally, the model accuracy increases linearly with respect to number of in-context examples before the phase transition but does not increase after the phase transition. The authors also analyzed the attention ratios between the task description and in-context examples, the impact on adding a loss in which the task is predicted, and then checking if their results generalize to a more common natural language dataset (CoFe).

I think this is a very technically solid work that could mainly be improved by improving the clarity (see my suggestions). I've rated the paper a 6 for now, but would be happy to re-evaluate the score if the clarity is improved.

**Strengths:**

* Good careful and methodical experimental design that combines both synthetic, controlled experiments and checks findings on a larger dataset.

*  The work employs multiple experimental measures/approaches that are intuitive and effectively support their hypotheses.

**Weaknesses:**

* Clarity: For Figure 3A, is the heatmap reflective of performance profile when adding the task prediction loss? The caption led me to believe that I'd be comparing two sets of data, one with task prediction loss and one without.

* Clarity: It seems Figure 3B would be better suited to be a part of Figure 2 and 3A for Figure 4 (if it is indeed a figure representing performance profile after adding the task prediction loss). Maybe this would make the figures have too many panels or introduce size constraints, but I encourage the authors to think about how each figure can have a consistent theme with panels that are related and support each other.

* Clarity: For Figure 4, the y-axis is the accuracy in predicting the task, right? If so, maybe label it as "task prediction accuracy." I had some brief confusion on whether this meant accuracy in performing the current task correctly or predicting what the current task is. Also would be good to label the color bar in Figure 4C.


* Interpretation: The interpretation of the task description suppressing the model's in-context learning ability puzzles me. To me, it seems like, when the task description does not have enough information, the model relies on the examples to learn the task. But when the task description has enough information, the model does not need to rely on examples because it has already learned the task, so additional examples don't lead to performance gains. In both cases, the task description is not necessarily hindering the model's in-context learning ability, but maybe is the "preferred" method of learning the task over examples. Maybe this is what the authors meant when they said "suppress", but to me, suppression of in-context learning would mean that the model is unable to learn the task using more in-context examples.

* Related to above: the results actually remind me of work in cognitive science on teaching with language vs demonstrations (Sumers et al. 2023). Teaching with language can be more effective because a language description of a task can effectively transmit abstract concepts needed to perform the task, whereas demonstrations give specific instances of concepts needed to perform the task and requires the learner to infer the abstract concepts given the demonstration. I'd be interested to hear the authors' thoughts on if their results can be interpreted in this light.

References:

Sumers, T. R., Ho, M. K., Hawkins, R. D., & Griffiths, T. L. (2023). Show or Tell? Exploring when (and why) teaching with language outperforms demonstration. Cognition, 232, 105326.

**Questions:**

* The authors mention real world scenarios as a limitation. What specifically about real world scenarios could make things different? It would be good to have specific examples. One I could think of: if a task requires grounding in another non-linguistic domain (e.g. utilizing images or motor programs for robots), then the task description can supply the abstractions needed to perform the task, but the in-context examples will be needed to ground those abstractions in the new domain. In this case, the learning dynamics after the phase transition may be different from the current results, because more in-context learning examples may still help the model even after the task description has enough information.

---

> ### Author Response · Authors · 2023-11-22
> **Rebuttal by Authors**
>
> Thank you for your valuable comments and suggestions and the encouraging words. We appreciate the time and effort you took to review our paper. Please find below our responses to your concerns and the changes we have made to address them.
>
> $\bullet$ Thank you for your feedback. We are regret that Figure 3A is not clear enough, and we have modified the figure and caption for clarity.
> The heatmap reflects the accuracy difference given two experiment settings: with or without task prediction. The "accuracy" here refers to the predicting accuracy of the answer of query equation, and "difference" means simple abstraction between accuracy values given same training data but under different training strategy. The two experiment settings use the same dataset, only differ in loss backward strategy. Only under task prediction setting, the loss between correct task and predicted task description will be backward. We will re-write the caption for a clearer expression.
>
> $\bullet$ We appreciate your suggestion. You are correct that figures need reorganization for consistent themes.
> We have made modifications to figures following your suggestion.
>
> $\bullet$ Thank you for your suggestion.
> The y-axis for plots in Figure 3B (Figure 4A in previous version) and Figure 3C (Figure 4B in previous version) refers to "task accuracy", which means accuracy in predicting what the current task is. We appreciate your insight and have made the necessary changes as per your suggestion.
>
> $\bullet$ Thank you for your valuable feedback. We are regret that our writing has led to your confusion.
> You are correct that the task description is more likely leading the model to ignore in-context examples. We have revisited and revised the paper to optimize interpretation.
>
> $\bullet$ Thank you for your suggestion and this is a very interesting point. Inspired by your comment, we have explored the relation between our results and  work in cognitive science on teaching with language vs demonstrations, and cited the related work.
>
> The experiments carried by Sumers et al.[1] are excellent reflection of in-context learning in real life. Teaching with language acts as adding task description, while teaching with demonstrations acts as giving in-context examples. The experimental results are very consistent with our synthesis experiments, that higher information of task description (teaching by language in a more informative and efficient way) will increase the lower bound of performance. And we offer theoretical derivation for this phenomenon in Eq.3. The phenomenon that language relies on shared abstractions to efficiently transmit complex concepts can also be illustrated by mutual information theory. As teaching languages concluding better shared abstractions are more informative and efficient, resulting in higher task description mutual info value.
>
> [1] Sumers, T. R., Ho, M. K., Hawkins, R. D., & Griffiths, T. L. (2023). Show or Tell? Exploring when (and why) teaching with language outperforms demonstration. Cognition, 232, 105326.
>
> $\textbf{Question}$: This is a very interesting point. Our experiments use simplified settings for mutual information quantization.
> One difference is that, in real world, the mutual information can be hard to calculated accurately, as the task description can be implicit and complex. We do believe the observation in this paper can inspire following works on the task description for in-context learning.

---

> > ### Comment · Reviewer_DYTw · 2023-11-22
> >
> > I appreciate the detailed revision. I've raised my score.
> >
> > Good luck to the authors!

---

### Official Review · Reviewer_YT61 · 2023-10-31

**Soundness:** 2 fair
**Presentation:** 3 good
**Contribution:** 2 fair
**Rating:** 5
**Confidence:** 3

**Summary:**

The authors investigate how task decripitions impact in-context learning (ICL) by conducting experiments on synthetic datasets. They find a phase transition in terms of the amount of information in task decriptions: (i) information below a threshold can hinder transformers from performing ICL; (ii) information above the threshold will promote ICL. Authors further try to interpret the phase (i) by showing that the task decriptions decrease the attention ratio of ICL examples. They also interpret phase (ii) by claiming rich task descriptions allow model to learn a good representation of the task. The authors also propose that using predicting the task label as an auxilliary task can improve ICL. Finally, the authors partially verify the previous discoveries on the real-world dataset CoFE.

**Strengths:**

1. I personally like the problem of how task information impacts ICL and I appreciate the idea of conducting synthetic experiments to approach the problem.

2. The illustrations are nice.

**Weaknesses:**

1. Some claims are not fully verfied or explained. (Will list them in the question section.)

2. I cannot see the difference between section 6 (Discussion) and section 8 (Conclusion).

Minor points:

3. The first paragraph on page 4 is irrelevant.

**Questions:**

1. For Figure 2A, why the blue curve achieves the best performance after the threshold?  In my understanding, blue curve stands for the least number of ICL expamples.

2. There seems to be 3 phases in Figure 2A so there should be 2 transitions. All the transition I mention in the comment is the latter one. I think the word transition refers different ones in different contexts, which makes me a bit confused. (For instance, the "phase transition" in the comment for Figure 2B in section 5.1 seems to be different from the transition in the 2nd paragraph in page 2.)

3. I don't see the reason behind the sharp transition in Figure 2A and the provided interpretations for both phase (i) and (ii) fail to explain that. Is the sharp transition caused by the lack of points used in the plot?


4. Phase (i) needs more detailed explaination as increasing the number of ICL examples barely improves ICL.

5. Section 5.2 gives the interpretation for phase (ii) by saying "Higher information of task description will increase the lower bound of performance." What is the lower bound here? Should it be stated in a more formal way?

6. The Figure 1C gives the i/o of the transformer model. Should the figure also include the output for the task prediction as eq(7) and eq(6) are using the same model $f$.

7. Is the first sentence in the 3rd paragraph of section 5.3 finished? ("As the task label prediction can also reflect whether the model understand what the task is.")

8. Figure 3A suggests that there is also a similar phase transition for the **accuracy gain**. Does it have any explaination?

9. Section 2 paragraph 2 says"However, despite these valuable contributions, all these explorations tend to overlook the influence of task descriptions on the in- context learning process. " Is that true? For the linear regression ICL papers, I don't think they have such task decriptions.

---

> ### Author Response · Authors · 2023-11-22
> **Rebuttal by Authors-part1**
>
> Thank you for your valuable comments and suggestions and the encouraging words. We appreciate the time and effort you took to review our paper. Please find below our responses to your concerns and the changes we have made to address them.
>
> $\bullet$ Thank you for your valuable feedback. We have revised corresponding sections for clarity and readability.
>
> $\textbf{Q1}$: Thank you for your question regarding Figure 2A. You are correct in your understanding that blue curve stands for the least number of ICL expamples. The phenomenon depicted in Fig 2A, that an increase in the number of ICL examples doesn't improve performance accordingly, is caused by relatively sparse sampling in in-context examples in Figure 2A. Please refer to Fig 2B, which demonstrates in-context examples' impact in continuous curves. We have listed the numerical results in the table below for clarity.
> It can be seen in Fig 2B that before the threshold (task info less than 3.2 nats), the number of in-context examples significantly impact in-context learning(the top one), while after that, it has almost no influence (the bottom one)}. And the bottom sub-figure in Fig 2B corresponding to the right most point in Fig 2A (i.e., mutual Info 4.02 nats), which clearly depicts that the accuracy stays relatively consistent across a large range of in-context example numbers. However, in-context examples do help as an accuracy gain can be observed, such as by increasing the in-context number from 0 to 4.
>
> |Accuracy (Mutual Info 0 nats) | 0.1017 | 0.1052 | 0.1117 | 0.1234 | 0.1320 | 0.2094 | 0.1876 | 0.2414 | 0.2955 | 0.3670 | 0.5367 |
> |----------------------------------------|-----------|-----------|----------|-----------|-----------|-----------|----------|-----------|-----------|-----------|-----------|
> |Accuracy (Mutual Info 4.02 nats) | 0.3530 | 0.4267 | 0.5052 | 0.4067 | 0.4695 | 0.4687 | 0.4656 | 0.4573 | 0.4678 | 0.4548 | 0.4674 |
>
> $\textbf{Q2}$: Thank you for your question about the use of word "transition". All "phase transition" in our work should refer to the latter one(i.e., the turning point at task info threshold around 3.2 nats), and we will check the presentation for clarity. The "phase transition" in Fig 2B also refers to the turning point at task info threshold around 3.2 nats.
> Special Attention is paid to this transition as it demonstrates a switch in task description's impact on in-context examples.
>
> $\textbf{Q3}$: Thank you for your question regarding the phase transition.
> first, We would also like to clarify that we have sampled 12 points in Figure 2A, and we have added the detailed value of mutual info and corresponding accuracy in the appendix. And there are around 3 points for phase (ii), and more than 8 points for phase (i). All these points with different number of in-context examples demonstrates the phase transition phenomenons. All these indicate that the reason behind the phase transition is not due to the lack of points used in the plot.
>
> It is an interesting point to study the phase transition phenomenons for in context learning, however it may be hard to provide an thorough explanation for the phase transition. For example, GROKKING[1] only presents the phase transition without explanation. And another work[2] observed a phase transition in investigating ICL given varying task diversity seen by pretrained model, and they suppose the sharp transition indicates a sharp crossover from the theoretical optimal of pre-trained task to that of true task.
>
> We would like to present some of our thoughts on the phase transition for discussion.
> We suppose the main reason behind this phase transition may related to the optimization.
> In phase (i), the optimization of the model is trapped into an ambiguous state when the model try to learn from both the context examples and insufficient task info.
> The task info, itself, can not provide sufficient guidance towards the right solution, and meanwhile distract the attention that paid on the examples. Even though the task info and the context examples are not inconsistent,  the model need to consider both factors, which may make it hard to be optimized. On the other hand, with sufficient task info in phase (ii), the optimization can mainly rely on the task info, leading to successful optimization and better performance.
>
> [1] Alethea Power, Yuri Burda, Harri Edwards, Igor Babuschkin, and Vedant Misra. Grokking: Generalization beyond overfitting on small al-gorithmic datasets, 2022
>
> [2] Allan Ravent ́os, Mansheej Paul, Feng Chen, and Surya Ganguli. Pretraining task diversity and the emergence of non-bayesian in-context learning for regression, 2023.

---

> ### Author Response · Authors · 2023-11-22
> **Rebuttal by Authors-part2**
>
> $\textbf{Q4}$: Thank you for your inquiry regarding Phase (i). We would like to clarify that increasing the number of ICL examples still improves model's performance, ad as an accuracy gain can be observed from in-context number 2 to in-context number 4 in Fig 2B. However, this improvement is soon interfered by high task info. Such phenomenon is mainly caused by ignorance to in-context examples led by task description, and can be illustrated more clearly by model's attention mechanism. As depicted in Fig 3B, a decline in the attention ratio of in-context examples can be observed when more task info added. This indicates that adding task description info will distract model's attention in in-context examples.
> In this phase, the accuracy stays relatively constant within a large range of in-context example numbers after threshold.
>
> $\textbf{Q5}$: Thank you for pointing out the omission of explanation of "lower bound". We regret that the interpretation here is not clear enough, and we have modified the text in the paper (highlighted in red). The interpretation of "lower bound" is formulated in Eq.3, and the "lower bound of performance" refers to the right part of the equation. The first term signifies the task label prediction, whereas the subsequent term corresponds to the loss function employed in the in-context training for the GPT model. This equation, therefore, demonstrates that accurate task label prediction contributes to the maximization of the log-likelihood. And this explanation matches the experimental results given in Fig 2 and Table 3 as under task label prediction setting, a performance gain can be observed with task info gain.
>
> $\textbf{Q6}$: Thank you for your suggestion, we have added the output for the task prediction in Figure 1C.
>
> $\textbf{Q7}$: We regret that the interpretation here is not clear enough, we have modified the text in the paper (highlighted in red) as follows:
> The performance of task label prediction can also reflect whether the model understand what the
> task is.
>
> $\textbf{Q8}$: Thank you for your question regarding Figure 4A (Figure 3A in the previous version). We would like to explain the phenomenon as follows:
>
> $\hspace{2em}$$\textbf{1. Figure 3A demonstrates a phase transition for the accuracy gain.}$ The x axis in Figure 3A refers to mutual info, while the y axis refers to number of in-context examples, exact labels have been added to the figure. The color of a point in Figure 3A refers to the accuracy difference with or without task prediction, given same number of in-context examples and same amount of task info. The accuracy gain is calculated as $A_{pred}-A_{nopred}$, where $A_{pred}$ stands for accuracy of results prediction with task prediction. The warmer the color, the larger the accuracy difference. As shown in Figure 3A, the accuracy gain increases sharply with mutual info, at a threshold around 3 nats (near the transition threshold demonstrated in Figure 2A).
>
> $\hspace{2em}$$\textbf{2. The phase transition is similar to that demonstrated in Figure 2A.}$ The accuracy gain increases sharply with mutual info, at a similar threshold with that in Figure 2A. Actually, both the accuracy of predicting task and w/o predicting task increase sharply around this threshold, as illustrated in Q3, caused by a crossover from learning from in-context examples to learning from task description.
>
>  $\hspace{2em} \textbf{3. The beneficial impact of predicting task label in in-context learning leads to the transition in accuracy difference.}$ As illustrated in Figure 3A, predicting task label will improve accuracy as the point are mainly colored warm. This indicates an improvement in understanding task description, when predicting task label. When more attention is added to task description, the advantage of task prediction will be amplified, resulting in a sharp increase in accuracy difference.
>
> $\textbf{Q9}$: Thank you for your valuable feedback. You are correct that linear regression ICL papers even don't have task descriptions. We regret that the expression here is inaccurate, and we have revised the text in the paper.

---

### Official Review · Reviewer_hvVV · 2023-11-01

**Soundness:** 2 fair
**Presentation:** 2 fair
**Contribution:** 3 good
**Rating:** 5
**Confidence:** 4

**Summary:**

**Update after rebuttal:**
I want to thank the authors for making many small improvements to the paper. Ideally, the paper would have been in this state when submitting already. Thank you also for running the ablations and pointing out that the task description is always helpful and never detrimental. To me personally the paper is approaching the acceptance threshold but still lies slightly below it. I have increased my score to reflect that. As I stated in my original review, I very much like the idea of creating synthetic tasks that have both in-context examples and a task-description that is "understood" by the model and can be modified in terms of its information content. I personally suggest to spend a bit more time to produce a strong and impactful paper that will stand the test of time - I think the potential for this is there. In the meantime, the current manuscript could be very suitable for an ICLR workshop for instance. Ultimately, I will not strongly argue against acceptance if the majority of reviewers thinks the paper is now ready, but to me personally the bar for ICLR has not quite been reached yet.

Should the paper get rejected, I personally think the most important areas to focus on are: more experimental evaluation (add one or two more tasks, increase the scale of tasks, think about ways to perform experiments at LLM scale (is there a way without having to train an LLM)), and working towards generality of the findings (do we get similar shapes for the usefulness of more task information across many tasks, or is the shape very task dependent?). Additionally, even with the many small improvements, the paper could use another pass to further improve the writing in terms of clarity and conciseness, and the figures are now OK but can also be polished a bit further.


**Summary:**
The paper investigates the role of two different sources of task information in in-context learning: (1) examples from the task (such as input-label pairs), and (2) general task descriptors (additional context that is informative about the task). The second type of information has received considerably less attention in the systematic analysis of in-context learning. The aim of this paper is to address this by designing a (relatively) small set of modular arithmetic tasks that are parameterized such that they are easily enumerable (leading to a unique task id, given their parameters). Additionally, a task-description can be constructed such that it provides a controllable (and quantifiable!) amount of information about the task parameters. This is implemented by providing lower and upper bounds for the task parameters as the task description; by widening or tightening the bounds, the amount of information provided can be easily manipulated. Taking all of this together allows to conduct precise experiments where the amount of task information available in the context can be controlled by: (1) the number of in-context examples, and (2) the amount of task information provided by the task description (which is prefixed in-context before the examples). The paper finds that providing no task information leads to the network performance depending strongly on the number of in-context examples, whereas providing full task information means accuracy is almost independent of the number of examples. Finally providing non-zero but less than (near-)maximal task information seems to be detrimental to in-context learning, leading to poor accuracy regardless of the number of examples in the context. Additionally the paper studies the influence of predicting the task id (the parameters of the task) as an auxiliary task on the previously obtained results; and performs an experiment inspired by the synthetic tasks using a non-synthetic LLM benchmark dataset.

**Strengths:**

* Very timely problem. Understanding the role of side-information for in-context learning is much less understood and studied compared to understanding the influence of in-context examples (particularly from a meta-learning / sequential prediction viewpoint).
* I really like the idea of constructing synthetic tasks where the amount of task-relevant side information in the context can be controlled and quantified precisely. This is a refreshingly different approach to yet another heuristic and ad-hoc method for improving in-context learning in an intransparent fashion.
* Forcing the network to explicitly predict the task might lead to the network “paying more attention” to the side information encoded in the task description in the context, and thus lead to improved overall performance. Testing this hypothesis is a really good idea.
* A limitations section.

**Weaknesses:**

* At the beginning of page 4 there seems to be a leftover comment from a co-author: “There are several grammar errors and issues with clarity in the given passage.”. Unfortunately I think that this applies to the whole paper except for Sections 1 and 2. The current paper is very cumbersome to chew through, and seems as if it was drafted and maybe reorganized multiple times in a hurry. Many conference papers have a paragraph here or there that is a bit rushed due to deadline pressure, but the current manuscript needs a major revision and rewrite in terms of clarity and readability. I personally suggest in the future to not give in too much to deadline pressure, if a paper is clearly not ready yet, take the time to polish it and turn it into the strongest possible version. This is doubly unfortunate because I really like the idea behind the main experiment; but it is currently not in a presentable state. I have left many concrete passages for improvement under ‘Questions’.
* Besides clarity, I do have some concerns regarding the main experiment and its interpretation. For Figure 2: the task info (mutual info) is given in the task description by a lower and an upper bound. I assume these values can only be integers and the mutual info can thus only be changed in discrete steps. The results in Fig. 2 A, C look like the task description is only helpful for the setting with maximal mutual information (which means the upper and lower bounds are equal), but is not helpful in any other setting. The main text describing the results of the figure says “after this information threshold the accuracy grows rapidly with information gain“ and the paper suggests similar, somewhat “gradual”, results on multiple occasions: it looks to me that this is incorrect; if the information in the task description is not maximal it seems to be detrimental regarding of the amount of information, and only when it is maximal does it actually help (i.e. only for a single value of the x-axis in Fig 2A, corresponding to the rightmost column in Fig 2C). Is this correct? If yes, I think this needs an explanation (why can the network not use partial info about the task; is this a general phenomenon or is it only for modular arithmetic, etc.).
* Figure 2: what are the maximally possible values for the lower and upper bound in the task description? Is it possible that for minimal task information the intervals are so large that they always have the same value (and the network easily learns to ignore these constant values in its input)? Whereas for non-zero but less than maximal task information these values change and act as detrimental noise that interferes with in-context learning? If yes, this could explain why for no task information we see non-trivial accuracy, but the kind of interference that we then see for non-zero task information seems less mysterious.
* Important ablations are missing:
    * No task description during training, i.e. only in-context examples. Gives a baseline performance.
    * No in-context examples, only task description with maximal info during training (sanity check: should lead to non-trivial accuracy, otherwise the network seems to be unable to use the task description).
    * No in-context examples, only task descriptions with low to medium task info during training (sanity check: should lead to worse but not detrimental performance; otherwise it indicates that the network cannot use any non-maximal-info task description and a redesign of the task is required).


**Verdict:** I was very excited to read this paper after the intro. The idea of controlling and quantifying the side-information in the context and study the impact on in-context learning is really good and interesting. The ablation with learning the task via an auxiliary loss is also very interesting. Unfortunately the paper in its current state does not hold up to its promises. The paper overall feels rushed and is a tedious read. Some important questions regarding the experimental setup are open (see questions below). Finally there are some concerns that the results shown do not quite match the interpretations in the paper (see weaknesses above). It is still unclear to me whether the results are a bit surprising (non-zero, but non-maximal task info seems to be very detrimental to in-context learning) or whether there is a simpler explanation that is harder to see because the task description only influences a single datapoint (when it is maximal). This needs to be cleared up, and requires some further investigation and ablations. While it is possible to tighten up the paper in terms of clarity and writing, I would suggest to come up with another synthetic task where task info can be manipulated even more fine-grained (some sort of injecting more or less noise into the description) to see whether the “phase transition” is generally this sharp or whether this is a quirk of modular arithmetic. Giving an answer to this would make for a potentially very strong paper. Overall I do not think that the current paper is ready for publication, but I want to encourage the authors to pursue what I believe to be a great main idea. I am happy to read the other reviews and hear the authors’ response before reaching a final conclusion.

**Questions:**

* Last paragraph on P1: point out what x, y, and r is.

* Notation: for a typical in-context set of examples $\{x_i, y_i, r_i \}_{i=1}^l, I assume that $r_i = r ~\forall i$, meaning all in-context examples have the same task index? After reading 4.2, I now understand that x and y are “inputs” and r is the “label” and not the task index. This makes sense, but really needs to be clarified in the intro.

* Typo: a few times in the paper: “the in-context learning” -> “in-context learning”, e.g. understanding in-context learning, or driving in-context learning.

* What is $r_a, r_b$ in Fig 1B?

* The Transformer used (24 layers, 8 attention heads) seems fairly large (or rather deep) for this task, how was the architecture chosen?

* I believe the index for the sum in Eq. 6 should start at i=2? Otherwise for i=1 the same x,y,r pair is used twice. Essentially the same for Eq. 7.

* End of Section 4.2: “We train the model for 20w steps” - 20 million steps?

* Is there any tokenization for the modular arithmetic task? If yes, which one, and does it ensure that two numbers from different parameters (e.g. a, and b in the task description) are never grouped into a single token?

* Fig. 2 needs improvements: What are the units on the x-axis (bits? nats?)? What are the thin colored lines in 2A (number of examples?)? Panel B: better to state the value of task info in addition to saying before and after phase transition. Panel B and C: replace “Example number” with “Number of examples”. Panel C: units on the x axis, numerical values on x- and y-axis ticks.

* **Important:** How is the network trained for the results in Fig. 2? Does the number of examples and amount of task information vary across training for a single model, or is a single model trained for one value of task information and one value of number of examples?

* Page 6, paragraph titled “Higher information of task description will increase the lower bound of performance.” - what figure/result does this paragraph refer to? It seems to be left over from a figure that has been dropped?

* **Important:** Page 7: how exactly was the extra loss for task prediction added? Did the model have to predict both the answer to the query and the correct task (otherwise how could the accuracy gain in Fig 3A be computed)? If yes, how were the two loss functions combined / weighted?

* Fig 3 needs similar improvement in terms of labeling as Fig 2.

* It took me literally 3 minutes to understand what data Fig 3A is showing. It seemed so unrelated to its description in the main text that I first thought that by accident the wrong figure ended up in the main text. The colorbar needs a label with a quantity that is mathematically defined in the main text. The plot itself needs an informative title.

* Fig 4 needs similar improvements in terms of labeling as Fig 2 and 3. Axes need units, axes ticks need numerical values, plots need informative titles, and legends need to say what variable the colors indicate - the list of logarithms in Fig 4B is incomprehensible. Colorbars need a label too to say what quantity they are encoding.

* **Important:** Essentially all line plots in the paper. Since lines show the mean over 5 runs, please also show some indication of variance (e.g. +/- std-deviation shaded areas). This is important for claiming significant difference between certain settings. Just because the means are different does not mean that the difference is statistically significant.

* **Important:** Sec. 5.4: I am not familiar with the CoFE dataset, and the current description in the paper does not really help with that. Please provide a better description (ideally with an illustration and examples) - if there is not enough space in the main paper, push it into the appendix. It is OK to leave out details and refer to the original publication, but readers should not need to read the publication to get the main gist of the dataset and tasks.

---

> ### Author Response · Authors · 2023-11-22
> **Rebuttal by Authors-part1**
>
> Thank you for your valuable comments and suggestions and the encouraging words. We appreciate the time and effort you took to review our paper. Please find below our responses to your concerns and the changes we have made to address them.
>
> $\bullet$ Thank you for your detailed feedback. We have thoroughly revised and polished the paper for clarity and readability.
>
> $\bullet$  We understand your concern about the main experiment. We would like to clarify that the medium task description also helps to improve performance.
>
> | Task Info(nats) | 0 | 0.21 | 0.45 | 0.73 | 1.02 | 1.39 | 1.61 | 1.83 | 2.30 | 3.22 | 3.40 | 3.91 | 4.02 | 4.27 | 4.61 |
> | ------ | ------ | ------ | ------ | ------ | ------ | ------ | ------ | ------ | ------ | ------ | ------ | ------ | ------ | ------ | ------ |
> |Accuracy(0 ex) | 0.1017 | 0.1020 | 0.1027 | 0.1030 | 0.1034 | 0.1030 | 0.1042 | 0.1048 | 0.1062 | 0.1119 | 0.1164 | 0.2104 | 0.2834| 0.4267 | 0.8641 |
> |Accuracy(4 ex) | 0.1118 | 0.1071 | 0.1062 | 0.1036 |  0.1023 | 0.1020 | 0.1030 | 0.1062 | 0.1083 | 0.1174 | 0.1202 | 0.2216 | 0.5052 |0.5877 | 0.9323 |
> |Accuracy(32 ex) | 0.3670 | 0.2083 | 0.1670 | 0.1046 | 0.1062 | 0.1070 | 0.1084 | 0.1097 | 0.1101 | 0.1211 | 0.1314 | 0.2137 | 0.4548 | 0.7926 | 0.9688 |
>
> We list the exact val accuracy under some experiment settings in Table above,  Accuracy(0 ex), Accuracy(4 ex), Accuracy(32 ex) refer to val accuracy of models trained with no in-context example, 4 in-context examples, 32 in-context examples. The results can provide a clearer demonstration of changes in accuracy, that accuracy basically grows with mutual info increasing. Before a certain information threshold (around 3.2 nats concerning results sampled in Table, the accuracy gain remains relatively subtle to be 0.001 between neighbouring info settings. And after this information threshold, the accuracy grows rapidly with information gain. The small accuracy increase before info threshold 3.2 nats is too small to be observed in Fig 2A and Fig 2C. Also, we actually sample several info value sampled after that information threshold (also see Table, referring to task description helps not only at maximal level. We are regret that our notations are not clear enough.
>
> We also explore the reason for model's relative low accuracy and subtle accuracy gain given under threshold task description, and find that the task description will lead the model to ignore the information from in-context examples. As depicted in Fig 2D, the ratio of in-context examples in attention keeps declining with more task description information. On the contrary, the attention ratio of task description increases when more task-related information are given. This indicates that adding task description info will distract model's attention in in-context examples.
>
> $\bullet$ Thank you for your detailed feedback and this is a very interesting point.
> The maximal possible value for upper bound in task description is 10, while the minimal possible value for lower bound is 1. Such interval seems not large enough to make task description change insignificant.
>
> And we would like to clarify that, according to Table above, task description is helpful whatever the amount of task info and the number of in-context examples given to the model. Especially when no in-context example is provided, the accuracy improves correspondingly with task info gain. Although the accuracy gain remains relatively subtle to be 0.001 before a certain information threshold (around 3.2 nats), an increase in task info does make beneficial improvement.
>
> As for the experiment phenomenon for non-zero task information, we look into the model's attention mechanism, and find that adding task description info will diverse model's attention in in-context examples. This shows that in-context learning is truly affected by adding task description, but in a continuous manner( See Fig 2D, the ratio of in-context examples in attention keeps declining with task info gain). The explanation may be the task description will lead the model to ignore the information from in-context examples.

---

> ### Author Response · Authors · 2023-11-22
> **Rebuttal by Authors-part2**
>
> $\bullet$ Thank you for your valuable suggestion, these ablation experiments are indeed important for verification of our results. We have added the following experiments as per your suggestion:
>
> $\hspace{1em}$ $\textbf{1. No task description}$: We present the accuracy given no task description and different number of in-context examples in the table below. It can be depicted that the accuracy grows with in-context example number. This table actually refers to zero mutual information in Figure 2A and Figure 2C, and it can be inferred from Figure 2 that model given full info task description always outperforms model given zero task info.
>
> |Number of In-context Examples | 0 | 4 | 8 | 12 | 16 | 24 | 32 | 36|
> | ------ | ------ | ------ | ------ | ------ | ------ | ------ | ------ | ------ |
> |Accuracy(0 ex) | 0.1017 | 0.1117 | 0.1234 | 0.1320 | 0.2094 | 0.2955 | 0.3670 | 0.5367 |
>
> $\hspace{1em}$ $\textbf{2. No in-context examples}$: Accuracy(0 ex) in Table given in point 2 lists the model's accuracy given different amount of task info and no in-context examples. When given maximal info (4.6052, referring to totally accurate task description), the model can achieve 0.8641 accuracy, better than all other info level settings, but fall behind models given both full task description and in-context examples. This infers model's ability in understanding task description. Also, it can be seen that under no example setting, the accuracy grows with information gain. The growing trend is relatively tiny given low task info, but speeds up when more task info added. Such performance pattern keeps align with experiments given both task description and in-context examples.
>
> $\bullet$$\textbf{A more fine-grained synthetic task:}$ Thank you for your valuable suggestion, and we fully appreciate your concern. We would like to clarify that our synthetic experiment setting can be more fine-grained by simply modifying the value range of $a,b$ in task description, a more detailed explanation are given as follows:
>
> $\textbf{1. The setting of task mutual information.}$ The mutual information we utilized here is formulated by $ln(n_{ab}/(r_a \cdot r_b))$, $r_a=a_u-a_l$ and $r_b=b_u-b_l$ stand for the inexact ranges of $a$ and $b$ are implied in task description, and $n_{ab}$ stands for the full number of available $ab$ pairs. Specifically, given maximal upper bound($a_u,b_u$) 10 and minimal lower bound($a_l,b_l$) 1 under the setting in the paper , there can be 100 different $r_a,r_b$ range pairs if $ab$ are restricted to integers.
>
> $\textbf{2. Enlarging the value range of $a$ and $b$ can improve granularity of task info.}$ By enlarging $a$ and $b$, the full number of available $ab$ pairs $n_{ab}$ is increased correspondingly. For example, given a larger maximal upper bound($a_u,b_u$) 100 with minimal lower bound($a_l,b_l$) 1, there can be 10000 different $r_a,r_b$ range pairs, resulting in 100 times more possible mutual info values.
>
> We have sampled 15 mutual info values in our experiment in the paper. Due to time constraints, we regret for not being able to present results with larger $a,b$ range here, but we promise to include results of more fine-grained experiment setting in final version.

---

> ### Author Response · Authors · 2023-11-22
> **Rebuttal by Authors-part3**
>
> $\textbf{Q1}$: Thank you for your suggestion, We have modified the corresponding parts in the paper(highlighted in red) as follows:
>
> The meta-learning framework is used to enrich in-context learning of Transformer, where the Transformer is directly trained to implement in-context learning. The task dataset for this framework is constructed by equations in the form of $(x \circ y)\ mod\ p = r$, where $p$ is a prime number, $\circ$ represents for operators, and $r$ is the result of equation to be predicted.Under this framework, the prompt is formulated as $[\{(x_i,y_i, r_i)\}_{i=1}^l,(x_q,y_q)]$.
>
> $\hspace{1em}$$\{(x_i,y_i,r_i)\}_{i=1}^l$ can be regarded as few shot examples, while $x_q$ is the validation examples.
>
> $\textbf{Q2}$: Thank you for your valuable suggestion, we have modified the text in the paper(highlighted in red).
>
> $\textbf{Q3}$: Thank you for your valuable feedback. We have revisited and revised the paper to unify presentation.
>
> $\textbf{Q4}$: We regret that our notation is not clear enough, we have modified the caption of Figure 1 for clarity.
> $r_a$ and $r_b$ are the inexact $ab$ ranges implied in task description. $r_a=a_u-a_l$, $r_b=b_u-b_l$,and $a_l$,$a_u$,$b_l$,$b_u$ stands for the possible lower and upper bounds of $a$ and $b$
>
> $\textbf{Q5}$: Thank you for your inquiry regarding model architecture.
>
> |Task Info(nats) | 0 | 0.21 | 0.45 | 0.73 | 1.39 | 1.83 | 3.22 | 4.02 | 4.27 | 4.61 |
> |-------------------- |---|-------|--------|--------|-------|--------|-------|--------|-------|--------|
> |Smaller Transformer(12 layers) | 0.1620 | 0.1034 | 0.1012 | 0.1004 | 0.1014 | 0.1014 | 0.1013 | 0.4507 | 0.771 | 0.9284 |
>
> We build our model following GPT2[1], choose to use a 24-layer transformer considering effectiveness of experiments. Performance of a smaller model with 12 layers are  shown in Table above. The models are given different amount of task info and 32 in-context examples. A smaller model still works and demonstrate similar phase transition in accuracy. However, compared to models under same experiment setting but trained with larger model(24 layer), a decline in val accuracy can be observed whatever task info given.
>
> A even smaller model (6 layer) gives worse performance, that even given exact task description and 32 in-context examples, the val accuracy is only 0.1258.
>
> To study how the task info affects the performance, we choose a larger model (24 layer) to better reflect the performance variation.
>
> [1] Ashish Vaswani, Noam Shazeer, Niki Parmar,Jakob Uszkoreit, Llion Jones, Aidan N Gomez, Ł ukasz Kaiser, and Illia Polosukhin. Attention is all you need. In I. Guyon, U. Von Luxburg, S. Bengio, H. Wallach, R. Fergus, S. Vishwanathan, and R. Garnett (eds.), Advances in Neural Information Processing Systems, volume 30. Curran Associates, Inc., 2017.
>
> $\textbf{Q6}$: Thank you for your inquiry regarding Eq.6 and Eq.7. We regret that our notation is not clear enough and we have re-formulated the loss function for clarification as follows:
>
> Given a token sequence $x=(x_1,\dots,x_T)$, we train the model to predict $p(x)=\prod_{t=1}^T p(x_t|x_{<t})$. We calculate loss for in-context examples, query and the answer of query equation. The in-context examples are denoted as set $\mathcal{C}_{i-1}$.
>
> For $i > 1$, $\mathcal{C}_{i-1}$
>
> $ =\{(x_1,y_1,r_1),\ldots,(x_{i-1},y_{i-1},r_{i-1})\}$
>
> For $i=1$, $\mathcal{C}_0$ is an empty set.
>
> Specifically, we calculate the loss for the sequence $s=\{(x_1,y_1,r_1),\ldots,(x_L,y_L,r_L)\}$ and task description $d$ as follows:
>
>    $ \mathcal{L}(\theta,s, d)=\frac{1}{L}\sum_{i=1}^L l( f(\{ d,\mathcal{C}_{i-1},x_i,y_i \}),r_i )$
>
> where $l$ denotes the loss function, e.g., cross entropy loss is adopted in our setting. The task description is $d = (a_l,a_u,b_l, b_u, op)$. Similarly, loss for task prediction (task $t=(a,b,op)$) can be re-formulated as:
>
> $ \mathcal{L}_t(\theta,s,d)$
>
> $= \frac{1}{L}\sum_{i=1}^L l( f(\{ d,\mathcal{C}_{i-1},x_i,y_i \}),r_i,t ). $

---

> ### Author Response · Authors · 2023-11-22
> **Rebuttal by Authors-part4**
>
> $\textbf{Q7}$: We apologize for this clerical error and here should be 200k steps. We have revised the text in the paper.
>
> $\textbf{Q8}$: Thank you for your questions regarding tokenization. We are sure numbers from different parameters will never be grouped into a single token.
> Following GROKKING[2], the only tokenization used in the modular arithmetic task is for operators. We represent $+$ as 13, $-$ as 16 and $/$ as 15, since $a$ and $b$ range from 1 to 10, and only the result will mod $p=11$, such tokenization ensures the operator won't be messed up with numerical parameters $a$, $b$ and result $r$. And as $a,b,r$ as well as lower bound/ upper bound all represent real numbers, there is no special need for extra tokenization.
>
> [2] Alethea Power, Yuri Burda, Harri Edwards, Igor Babuschkin, and Vedant Misra. Grokking:Generalization beyond overfitting on small al-gorithmic datasets, 2022.
>
> $\textbf{Q9, Q13,Q14,Q15,Q16}$: Thank you for your suggestion. We have modified all the plots, adding axis, labels, captions and variance for clarity.
>
> $\textbf{Q10}$: Thank you for pointing out the omission of network training strategy in the paper. A single model is trained with one fixed value of task info as well as one fixed value of number of in-context values.
>
> $\textbf{Q11}$: Thank you for your question regarding "the lower bound of performance".
> As given in Eq.3 in the paper, the "lower bound of performance" refers to the right part of the equation. The first term signifies the task label prediction, whereas the subsequent term corresponds to the loss function employed in the in-context training for the GPT model. This equation, therefore, demonstrates that accurate task label prediction contributes to the maximization of the log-likelihood. And this explanation matches the experimental results given in Fig 2 and Table given in Rebuttal Part 1, as under task label prediction setting, a performance gain can be observed with task info gain.
> We are regret that our presentation is not clear enough and we have modified the text in the paper.
>
> $\textbf{Q12}$: Thank you for your inquiry regarding task prediction loss.
> The loss function for task prediction is formulated as Eq. 7. Compared to Eq. 6, we add the correct task description to the end of token sequence to supervise task prediction. As we use auto-regression strategy, the information of accurate result $r_i$ and real task description $t$ won't disclose and kept invisible to model during prediction. So "added" actually means "added to the ending of input token sequence", rather than manipulation on loss.
>
> $\textbf{Q17}$: Thank you for your valuable feedback. We have added detailed description of CoFE and several examples in the appendix.

---

### Author Response · Authors · 2023-11-22
**Official Comment by Authors**

Thanks for AC and reviewers spending time in handling this paper. We thank all the reviewers for the positive feedback and constructive comments, as well as their encouraging words: The paper studies a $\textbf{very timely problem} $(Reviewer hvVV). The problem of how task information impacts ICL is liked (Reviewer YT61). The idea of conducting synthetic experiments to approach the problem is $\textbf{appreciated}$ (Reviewer YT61), and is $\textbf{a refreshingly different approach}$ to yet another heuristic and ad-hoc method for improving in-context learning in an intransparent fashion (Reviewer hvVV). The experimental design is $\textbf{careful and methodical}$ that combines both synthetic, controlled experiments and checks findings on a larger dataset(Reviewer DYTw). The proposed task and experimental setup were $\textbf{clean}$ (Reviewer 9ENX). The work employs multiple experimental measures/approaches that are $\textbf{intuitive and effectively support their hypotheses}$ (Reviewer DYTw).

The improvements in the main draft can be listed as (highlighted in red):

1. As suggested by Reviewer hvVV, we added ablation experiments in the paper.

2. As suggested by Reviewer hvVV and Reviewer 9ENX, we unified and supplemented the notations.

3. As suggested by Reviewer hvVV, Reviewer DYTw and Reviewer 9ENX, we revised the figures in the paper, adding labels for axis and colorbars.

4. As suggested by Reviewer hvVV, we added shaded areas as indication of variance in Figure 2 and Figure 4.

5. As suggested by Reviewer hvVV, we improve the notations in Eq. 6 and Eq. 7 to avoid misunderstanding.

6. As suggested by Reviewer hvVV and Reviewer YT61, we revised the writing in section 5.2 to avoid confusion in "lower bound of performance".

7. As suggested by Reviewer hvVV, we added detailed explanation of CoFE and related examples in appendix A.4

8. As suggested by Reviewer hvVV and Reviewer YT61, we provided a more detailed interpretation for Figure 2 in appendix A.3, numerical results of points used in plots are also listed.

9. As suggested by Reviewer hvVV and Reviewer YT61, we revised Figure 1, improved the notations and added input/output for task prediction setting.

10. As suggested by Reviewer YT61, we revised the related work section and modified our writing to avoid misunderstanding.

11. As suggested by Reviewer hvVV, Reviewer YT61 and Reviewer 9ENX, we removed the unrelated paragraph on page 4.

12. As suggested by Reviewer YT61, we revised the first sentence in the 3rd paragraph of section 5.3.

13. As suggested by Reviewer DYTw, we re-organized the order of plots for better demonstration.

14. As suggested by Reviewer DYTw, we revised our wording, using interprations like "the task description will lead the model to ignore the information from in-context examples" instead of "suppression" or "hinder".

15. As suggested by Reviewer DYTw, we explored the relation between our results and work in cognitive science on teaching with language vs demonstrations, and added corresponding related work.

16. As suggested by Reviewer hvVV and Reviewer 9ENX, we revisited the paper and checked ungrammatical writing.

17. As suggested by Reviewer 9ENX, we opted to use "transformers" instead of "LLMs" to avoid misunderstanding.

18. As suggested by Reviewer YT61, we reorganized discussion section and conclusion section.

---

### Author Response · Authors · 2023-11-23

Thank you for all of your constructive comments and suggestions. Please let us know as soon as possible if you have any further questions or concerns, since the discussion stage is coming to an end soon.

---

### Meta-Review · Area_Chair_nuGs · 2023-12-09

**Metareview:**

While the paper received diverse scores (5, 5, 8, and 3), the majority of reviewers agreed that it is not currently strong enough for publication at this conference. However, the reviewers also acknowledged the exciting potential of the research direction, particularly highlighting the intriguing "phase transition" finding. With significant improvements in clarity, presentation, and methodological rigor, the paper could develop into a valuable contribution to the field and be considered for future publication.

One of the key weaknesses identified by reviewers was the lack of clarity. The paper was described as being difficult to follow, with issues in flow and formatting errors, suggesting that it was rushed or incomplete. Although additional experiments and clarifications were added during the revision period, it was agreed during the course of reviewer discussions that more formal and quantitative investigation was needed on a wider variety of tasks, perhaps at larger scales, in order to verify this “phase transition” finding.

I would like to commend reviewer hvVV for initiating a valuable discussion among all reviewers and providing insightful feedback for the authors. I encourage the authors to carefully consider their suggestions for further improvement.

**Justification For Why Not Higher Score:**

Given the lack of clarity in the presentation as well as the need for a more comprehensive set of experiments, I don't think it's justified to overrule the decision of 3 out of 4 reviewers to reject.

**Justification For Why Not Lower Score:**

NA

---

### Decision · Program_Chairs · 2024-01-16

Reject